# Crochet Hook Technique for Arthroscopic Anterior Talofibular Ligament Repair: Technique Note

**DOI:** 10.3390/jcm11236922

**Published:** 2022-11-24

**Authors:** Zitao Liu, Jing Li, Gengxin Chen, Shihua Gao, Enhui Feng, Haitao Su, Haiyun Chen, Tao Jiang

**Affiliations:** 1Department of Orthopaedics, The Second Affiliated Hospital of Guangzhou University of Chinese Medicine, Guangzhou 510261, China; zitaoliu@gzucm.edu.cn (Z.L.); chengengxin3495@gzucm.edu.cn (G.C.); feh@gzucm.edu.cn (E.F.); haitaosu@gzucm.edu.cn (H.S.); 2The Second Clinical School of Guangzhou University of Chinese Medicine, Guangzhou 510006, China; 20201120378@stu.gzucm.edu.cn (J.L.); 20202120037@stu.gzucm.edu.cn (S.G.)

**Keywords:** crochet hook, anterior talofibular ligament, arthroscopic repair, lateral instability of the ankle, ankle arthroscopy

## Abstract

Ankle sprains can lead to chronic lateral ankle instability caused by an injured anterior talofibular ligament (ATFL), and surgery is often required when conservative treatments fail. BROSTROM surgery is considered the gold standard and has a definite curative effect. Advancements in arthroscopic surgery and improvements in implanted anchors have led to an increase in ATFL repairs using arthroscopic surgery. Arthroscopic AFTL repair is less invasive, and patients could experience faster recovery compared to open AFTL repair. To simplify the complicated suture-passing processes in arthroscopic AFTL repair, we developed a crochet hook and loop wire technique, which is described in this paper.

## 1. Introduction

Chronic ankle instability is a common musculoskeletal problem often associated with the anterior talofibular ligament (ATFL) [1,2,3,4]. Conservatively dealing with most ATFL injuries requires more prolonged rehabilitation to return to normal activities [5,6,7]. When conservative measures fail to improve functional ankle instability, ATFL reconstruction is required to prevent post-traumatic sequelae, such as osteochondral lesions of the talar dome and ankle osteoarthritis [8].

Previous reports have shown satisfactory clinical results in treating chronic ankle instability using the standard open Bostrom procedure for ATFL repair. However, complications still occur in 6% to 25% of cases, including ankle pain, swelling, joint stiffness, and the recurrence of ankle instability [9,10,11]. The arthroscopic repair of ATFL becomes a better option, which is less invasive than open surgery, reduces postoperative pain, and enables faster recovery [3,12,13,14,15,16,17,18]. Therefore, the arthroscopic repair of ATFL will become popular in treating chronic lateral ankle instability.

At present, there have been published several technical reports on repairing ATFL under arthroscopy, but without exception, complex suture-passing operations are required. The aim of this technical note is to introduce a method of suture-passing operations to improve the efficiency of surgery.

## 2. Technical Note

The patient was placed in a supine position with the ankle in 90-degree dorsiflexion after spinal anesthesia. A 5.0 mm/30° arthroscope (Arthrex, Naples, FL, USA) and a 2.3 mm ablator (DePuy Orthopedics, Warsaw, IN, USA) were used for arthroscopic procedures. The crochet needle (Xigege Co., Ltd., Zhe Jiang, China) was made of stainless steel, the rear handle was about 90 mm long, and the diameter was about 10 mm. The front needle was about 60 mm long, the needle diameter was about 1 mm, and the needle body diameter was about 1.5 mm (Figure 1).

During the procedures, two portals were created: an anterior medial (AM) portal and an anterior lateral (AL) portal (Appendix A). The AM portal was located at the medial edge of the anterior tibialis tendon (ATT), and an incision length of about 8 mm was made distally parallel to the tip of the medial malleolus (MM). The AL portal was located lateral to the third peroneal muscle (TPM), at the same level as the AL portal (Figure 2). To avoid nerve damage, the AL portal was used, which involved incising the skin with the blade and then blunt dissection using a mosquito clamp before accessing the joint cavity. When entering the joint through the AM portal, we performed an arthroscopy first, including soft tissue impingement, bony hyperplasia, loose bodies, talus cartilage injury, and targeted treatments such as synovial debridement, microfracture, or osteophyte wear arthroplasty. At the same time, the ATFL and stump quality were assessed. When it was clear that the ATFL tear and the quality of the stump could be repairable, a fully arthroscopic anatomical repair could be performed.

A suture anchor (DePuy Orthopedics, Warsaw, IN, USA) was placed on the fibular footprint of the AFTL through the AL portal. About 5 mm from the edge of the articular cartilage, where the anchor was located, was the anatomical footprint of the ATFL (Figure 3). After determining the location, a bone grinder was used to grind the bone bed. The Basset’s ligament may have been worn due to impingement from the anterolateral edge of the talus. To prevent iatrogenic injury to the articular surface when drilling the hole for the suture anchor, the sleeve drill needed to be parallel to the lateral surface of the talus while the handle was tilted to the distal end. The sleeve drill was drilled from anterior to posterior to prevent the anchor from penetrating behind the lateral malleolus and irritating the peroneal tendon. Then, holes were drilled to implant anchors.

A loop wire was formed on the exterior of the ATFL by piercing the skin and entering the ATFL under supervision. Then, the surgeon grasped the suture at one end and dragged it out beneath the skin. A mosquito clamp was inserted through the AL portal, a little bit of the loop wire between the ATFL and the lateral joint capsule was withdrawn, and then, the mosquito clamp was pulled through the loop wire, and the other end of the suture was clamped out of the joint through the AL portal (Figure 4). This is the crochet hook technique. At this time, the ATFL was covered by the suture loop. The suture was tightened to make the ATFL close to the stop point of the LM. A knot pusher could be utilized to tie the knot in the AL portal, and the suture was then cut (Figure 5).

## 3. Discussion

The suture-passing operations discussed in this paper have several advantages. First, the crochet hook technique simplifies operations and penetrates the ligament to form the suture loop, which can save a significant amount of time. Secondly, the contralateral suture is directly drawn in the joint, which simplifies complex in vitro suture routing, avoids the situation of suture wrapping, and improves surgical efficiency. Third, the knot is fixed in the joint to avoid skin irritation. One methodical limitation is that crochet tools are thin and fragile. Overly large operation angles can result in breakages and foreign matter residue. To prevent fatigue breakage, each crochet tool should be used once.

This technique is suitable for patients with good stump quality after ATFL injury. Contraindications include stump degeneration or absorption that cannot be repaired. In patients with lateral malleolus avulsions, anchor implantation may be difficult due to the size and location of the bone fragments, as well as the operator’s experience.

## Figures and Tables

**Figure 1 jcm-11-06922-f001:**
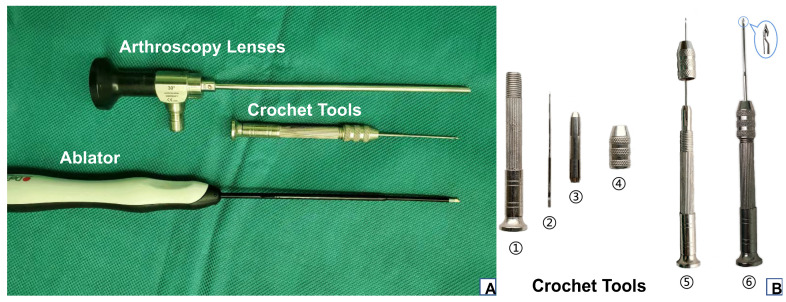
(**A**) Operative instrument. (**B**) An overview of crochet tool’s composition and local details: (①) hollow handle, (②) crochet hook, (③) tongs, (④) tightening nut, (⑤) crochet hook assembly diagram, (⑥) enlarged view of the crochet hook head end.

**Figure 2 jcm-11-06922-f002:**
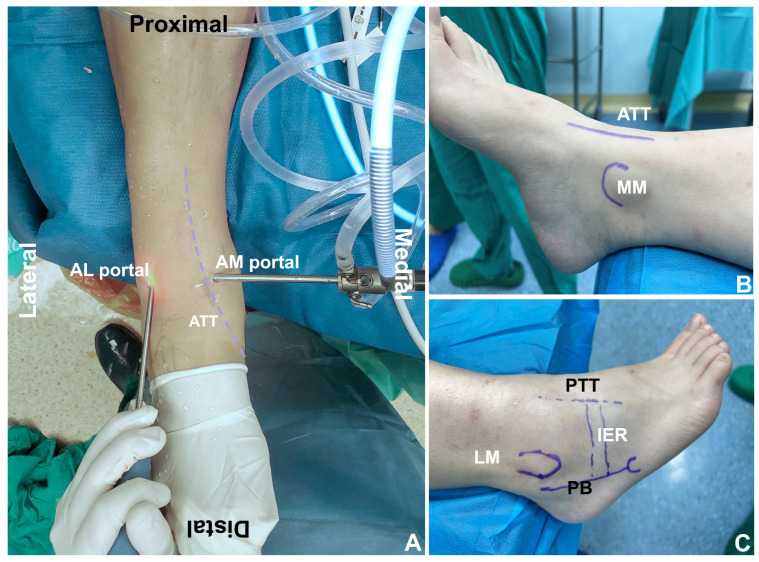
Portal placement and anatomic landmarks. (**A**) Anterior view. Indicate the anterior medial (AM) and anterior lateral (AL) portals. ATT: anterior tibial tendon. (**B**) Medial view. MM: medial malleolus. (**C**) Lateral view. PTT: peroneus tertius tendon, PB: peroneus brevis, IER: inferior extensor retinaculum, LM: lateral malleolus.

**Figure 3 jcm-11-06922-f003:**
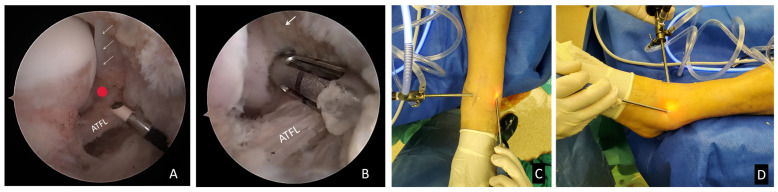
(**A**) The ATFL fibular footprint (red dot) is located approximately 5 mm inferior to the distal bundle of the anterior inferior tibiofibular syndesmosis (white arrow). (**B**) An anchor is placed in the footprint. (**C**) Top view, the direction of the anchor is parallel to the lateral surface of the talus. (**D**) Lateral view, with the anchor oriented at an approximately 45-degree angle to the lateral malleolus.

**Figure 4 jcm-11-06922-f004:**
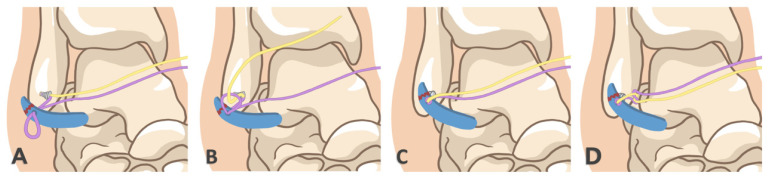
Loop principle: (**A**) Pass the purple wire with the anchor nail through the ligament to form a loop on the opposite side. (**B**) Pass the yellow wire with the anchor nail through the loop. (**C**) Tighten the purple wire through the ligament and then the yellow wire to wrap the ligament and pull it towards the bony surface. (**D**) Finally, the ligament repair is completed by knotting and fixing.

**Figure 5 jcm-11-06922-f005:**
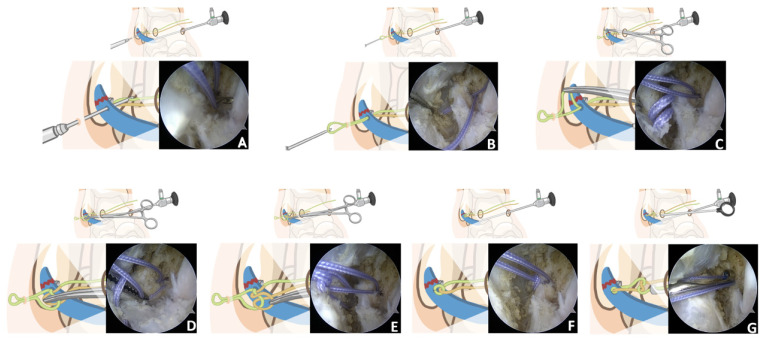
(**A**) The suture is ready for withdrawal once the crochet hook is passed through the ATFL. (**B**) The crochet hook is then used to create a loop wire. (**C**) Mosquito clamp is inserted through AL portal, and the loop wire is clamped out. (**D**) Grasping the contralateral suture, mosquito clamp passes through the loop wire. (**E**) Clamping out the contralateral suture. (**F**) After tightening the sutures, the ATFL is pulled towards the point of attachment. (**G**) Knot pushers are used to fix the ligament in place.

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
