# Peer review of "Crochet Hook Technique for Arthroscopic Anterior Talofibular Ligament Repair: Technique Note"

_jcm, 2022, doi:10.3390/jcm11236922_

Round 1
Reviewer 1 Report
I had the pleasure to read a technical note on a surgical technique for ATLF reconstruction. The English level, spelling and expressions in the manuscript should definitely be improved, as they seem to be translated from a different language.
Row 23 - replace the word "cannot" with a different similar synonym
Row 31 - Do not use capital letter for Arthroscopic
Figure 1 - please be more specific with the legend. "Tools" is not a mention that suggests what we see in the image.
Please specify what A and B means in this figure.
Row 56 - Arthroscopy is entered - this does not make any sense in English. Please rephrase.
Figure 5 - The highlight of the "A" and "a" cannot be differentiated in the figure. Please make it clearer.
Please provide a conclusion section with shortened recommended tips and tricks for the technique.
References are correctly written.
Author Response
Thank you for your consideration.

Reviewer 2 Report
Thanks to the Editor for inviting me to review this interesting paper. The authors have made a great effort to describe their suture technique on anterior tibiofibular ligament arthroscopic repair, and I endorse its publication after major revision.
OVERALL
The manuscript is well-written and concise and depicts a novel and easy suture technique for the anterior tibiofibular ligament arthroscopic repair, a common procedure in current sports orthopaedic surgery. However, adding a “discussion” subheading will be valuable for commenting on the benefits of this technique and contrasting it with the already reported ones.
STRENGTHS
Concise.
Elaborated description of the surgical technique.
Fantastic images.
WEAKNESSES
One of the images is too small to allow the proper visualization.
Lacks a proper discussion highlighting the technique’s benefits and contrasting it with other anterior tibiofibular ligament suture techniques.
SECTION BY SECTION
Introduction:
It is concise, with adequate referencing.
Please, state the objective of the manuscript at the end of the introduction.
Lines 35-40, starting with “The suture-passing…” should be moved to a “Discussion” section.
Technical note:
Reword the first paragraph to: “The patient was placed in a supine position with the ankle in 90-degree dorsiflexion after spinal anesthesia. A 5.0 mm/30° arthroscope (Arthrex) and a 2.3 mm ablator were used for arthroscopic procedures. The crochet needle is made of stainless steel, the rear handle is about 90 mm long, and the diameter is about 10 mm. The front needle is about 60 mm long, the needle diameter is about 1 mm, and the needle body diameter is about 1.5 mm (Figure 1).
Is the crochet tool a surgical instrument? Or is it an instrument whose function was translated for this surgery? Please, add a couple of sentences in this regard.
Line 60. Substitute the word detected for: assessed or evaluated.
Line 70. Substitute insertion point for “footprint”.
Line 71. Please, correct Basset´s ligament.
Lines 87 and 88 show an inadequate abbreviation for the arthroscopic portals, which may confuse them. Please, address it accordingly.
Please, add a “Discussion” subheading to discuss: (1) the benefits and disadvantages of the crochet suture technique; (2) Other suture techniques, including the standard one; and (3) compare them. Conclude by highlighting what this technique brings to the current ligament repair technique and when we should use it.
References:
Adequate and up-to-date.
Figures:
Figure 2:
The image identified as “A” can be portrayed vertically, giving the surgeon a better visualization. The other pictures can be fitted on the right side of that image.
Image C peroneus tertius tendon abbreviation does not match the one in the legend.
Correct IER: inferior extensor retinaculum.
Figure 3. Consider splitting the images into many, as the pictures need to be visualized adequately. It is worth it to have such a pictorial description.
Abstract and title:
Adequate.
Author Response
Thank you For your consideration.

Round 2
Reviewer 1 Report
The requested changes are made and the article has a technical view now.
Reviewer 2 Report
Congratulations to the authors for the innovative suture-passing technique for arthroscopic ATFL repair. I hope the readers enjoy this simple but time-saving technique as much as I did when reviewing the manuscript.
All comments and queries were answered satisfactorily, and the manuscript can proceed to the following editorial steps in the rest of the reviewers agree to it.